# Neural Implicit Representations for Physical Parameter Inference from a Single Video

## Abstract

Neural networks have recently been used to model the dynamics of diverse physical systems. While existing methods achieve impressive results, they are limited by their strong demand for training data and their weak generalization abilities. To overcome these limitations, in this work we propose to combine neural implicit representations for appearance modeling with neural ordinary differential equations (ODEs) in order to obtain interpretable physical models directly from visual observations. Our proposed model combines several unique advantages: (i) Contrary to existing approaches that require large training datasets, we are able to identify physical parameters from only a single video (ii) The use of neural implicit representations enables the processing of high-resolution videos and the synthesis of photo-realistic imagery. (iii) The embedded neural ODE has a known parametric form that allows for the identification of interpretable physical parameters, and (iv) long-term prediction in state space. (v) Furthermore, the photo-realistic rendering of novel scenes with modified physical parameters becomes possible.

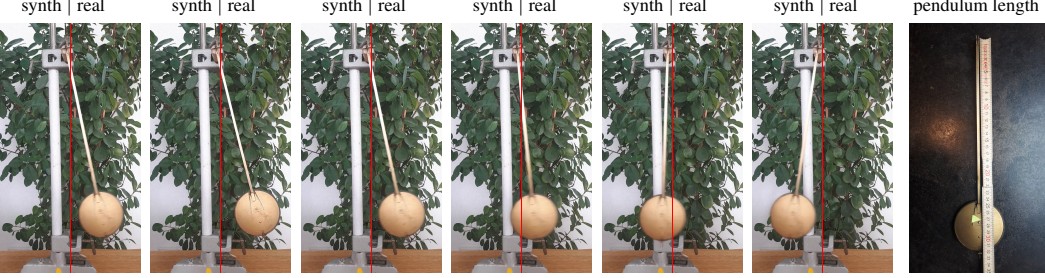

Figure 1: Our method infers physical parameters directly from real-world videos, like the shown pendulum motion. Separated by the red line, the right half of each image shows the input frame, and the left half shows our reconstruction based on physical parameters that we estimate from the input. We show 6 out of 10 frames that were used for training. The proposed model can precisely recover the metric length of the pendulum from the monocular video (relative error to true length is less than 2.5%). Best viewed on screen with magnification. Please also consider the supplementary video.

## 1 Introduction

The physics of many real-world phenomena can be described concisely and accurately using differential equations. However, such equations are usually formulated in terms of highly abstracted quantities that are typically not directly observable using commodity sensors, such as cameras. For example, a pendulum is physically described by the deflection angle, the angular velocity, the damping coefficient, and the pendulum's length, but automatically extracting those physical parameters directly from video data is challenging. Thus, due to the complex relationship between the physical process and images of respective scenes, measuring such quantities often necessitates a trained expert operating customised measuring equipment. While for many physical phenomena humans are able to infer (a rough estimation of) physical quantities from a given video, physical understanding from videos is an open problem in machine learning.

Recently, the combination of deep learning and physics has become popular, particularly in the context of video prediction. While earlier works (Lutter et al., 2019; Greydanus et al., 2019; Cranmer et al., 2020; Zhong et al., 2020) require coordinate data, i.e. already abstracted physical quantities, more recent works directly use image data (Levine et al., 2020; Zhong & Leonard, 2020). A major downside is that all these approaches rely on massive amounts of training data, and, as we experimentally confirm in App. F, they exhibit poor generalization abilities. In contrast, in our work we address this shortcoming by proposing a solution that extracts semantic physical parameters directly from a single video, see Figure 1. Therefore, we alleviate the need for large data and furthermore facilitate interpretation due to the semantics of the inferred parameters in respective physical equations. Additionally, the six previously mentioned works model physical systems using Lagrangian or Hamiltonian energy formulations, which elegantly guarantee the conservation of energy, but can therefore not easily model dissipative systems that are much more common in the real world (Galley, 2013). The proposed model effectively transforms the camera into a physical measuring device with which we can observe quantities such as the length or the damping coefficient of a pendulum.

To achieve the learning of physical models from a single video, we propose to utilise physics-based neural implicit representations in an analysis-by-synthesis manner, where the latter relies on neural ordinary differential equations for representing abstract physics of visual scenes. Overall, we summarize our main contributions as follows:

1. We present the first method that is able to identify physical parameters from a single video using neural implicit representations.
2. Our approach infers parameters of an underlying ODE-based physical model that directly allows for interpretability and long-term predictions.
3. The unique combination of powerful neural implicit representations with rich physical models allows to synthesize high-resolution and photo-realistic imagery. Moreover, it enables physical editing by rendering novel scenes with modified physical parameters.
4. Contrary to existing learning-based approaches that require large corpora of training data, we propose a *per-scene* model, so that only a single short video clip that depicts the physical phenomenon is necessary.

## 2 RELATED WORK

The combination of machine learning and physics has been addressed across an extremely broad range of topics. For example, machine learning was used to aid physics research (Bogojeski et al., 2020; Leclerc et al., 2020), or physics was used within machine learning models, such as for automatic question answering from videos (Chen et al., 2021; Bear et al., 2021). In this work we focus specficially on extracting physical models from single videos, so that in the following we discuss related works that we consider most relevant in this context.

**Physics in the context of learning.** While neural networks have led to many remarkable results across diverse domains, the inference of physical principles, such as energy conservation, is still a major challenge and requires additional constraints. A general way to endow models with a physics-based prior is to use generalized energy functions. For example, Greydanus et al. (2019) and Toth et al. (2020) use a neural network to parameterize the Hamiltonian of a system, which yields a relation between the energy of the system and the change of the state. Hence, they are able to infer the dynamics of systems with conserved energy, such as a pendulum or a multi-body system.

One disadvantage of using the Hamiltionian is that *canonical coordinates* need to be used. To eliminate this constraint, other works use the Lagrangian to model the system's energy. Since this formalism is more complex, Lutter et al. (2019) and Zhong & Leonard (2020) restrict the Lagrangian to the case of rigid-body dynamics to model systems with multiple degrees of freedom, such as a pole on a cart, or a robotic arm. Cranmer et al. (2020) use a neural network to parameterize a general Lagrangian, which they use to infer the dynamics of a relativistic particle in a uniform potential.

While being able to model many relevant systems, the aforementioned energy-based approaches cannot easily be extended to dissipative systems that are much more common in the real world (Galley, 2013). Furthermore, they do not allow for a semantic interpretation of individual learned system parameters. PhyDNet, introduced by Guen & Thome (2020), learns dynamics in the form of a general PDE in a latent space, which, like the aforementioned works, prohibits interpretation

of the learned physical model. In contrast, in the context of incorporating physical phenomena into learning frameworks, there are also approaches that make the underlying dynamics explicit. For example, Jaques et al. (2020) unroll the Euler integration of the ordinary differential equation of bouncing balls, as well as balls connected by a spring, to identify the physical parameters like the spring constant. Kandukuri et al. (2020) and de Avila Belbute-Peres et al. (2018) propose to use a linear complementarity problem to differentiably simulate rigid multi-body dynamics that can also handle object interaction and friction. For our method, we also rely on the advantages of modelling the underlying physics explicitly in order to obtain interpretable parameter estimates.

**Inferring physical properties from video.**        While many approaches work with trajectories in state space, there are also some works that operate directly on videos. In this case, the information about physical quantities is substantially more abstract, so that uncovering non-linear dynamics from video data is a significantly more difficult problem. Traditionally, such inverse problems are often phrased in terms of optimization problems, for example for deformable physics inference (Weiss et al., 2020), among many more. While respective approaches can successfully estimate a wide range of relevant physical quantities from video data, they often require rich additional information, such as 3D information in the form of depth images in combination with a 3D template mesh (Weiss et al., 2020), which may limit their practical applicability.

More recently, several end-to-end learning approaches have been proposed. de Avila Belbute-Peres et al. (2018) use an encoder to extract the initial state of several objects from the combination of images, object masks and flow frames. After propagating the physical state over time, they decode the state back into images to allow for end-to-end training. Jaques et al. (2020) and Kandukuri et al. (2020) use an encoder network to extract object positions from object masks for individual frames. After estimating initial velocities from the positions they integrate the state over time and use a carefully crafted coordinate-consistent decoder, which is based on spatial transformers, to obtain predicted images. Zhong & Leonard (2020) extend this idea to their variational autoencoder (VAE) architecture to obtain a coordinate-aware encoder which they use to infer parameters of the latent distribution of generalized coordinates for each frame. Toth et al. (2020) use a VAE structure to predict the parameters of a posterior over the initial state from a sequence of videos. All of these approaches require large amounts of data to train the complex encoder and decoder modules. In contrast, our approach does not rely on trainable encoder or decoder structures, but instead uses a non-trainable fixed neural ODE solver in combination with a trainable neural implicit representation, and is thus able to infer physical models from a single video.

**Implicit representations.**        Recently, neural implicit representations have gained popularity due to their theoretical elegance and performance in novel view synthesis. The idea is to use a neural network to parametrize a function that maps a spatial location to a spatial feature. For example, to represent geometric shapes, using occupancy values (Mescheder et al., 2019; Chen & Zhang, 2019; Peng et al., 2020), or signed distance functions (Park et al., 2019; Gropp et al., 2020; Atzmon & Lipman, 2020). In the area of multiview 3D surface reconstruction as well as novel view synthesis, implicit geometry representations, such as density or signed distance, are combined with implicit color fields to represent shape and appearance (Sitzmann et al., 2019; Mildenhall et al., 2020; Yariv et al., 2020; Niemeyer et al., 2020; Azinovic et al., 2021). To model dynamic scenes, there have been several approaches that parametrize a displacement field and model the scene in a reference configuration (Niemeyer et al., 2019; Park et al., 2021; Pumarola et al., 2021). On the other hand, several approaches (Xian et al., 2021; Li et al., 2021; Du et al., 2021) include the time as an input to the neural representation and regularize the network using constraints based on appearance, geometry, and pre-trained depth or flow networks – however, none of these methods uses physics-based constraints, e.g. by enforcing Newtonian motion. While the majority of works on implicit representations focuses on shape, Sitzmann et al. (2020) show the generality of implicit representations by representing images and audio signals. Our work contributes to the neural implicit representation literature by combining such representations with explicit physical models.

## 3    ESTIMATING PHYSICAL MODELS WITH NEURAL IMPLICIT REPRESENTATIONS

Our main goal is the estimation of physical parameters from a single video, where we specifically focus on the setting of a static background and dynamic objects that are moving according to some

physical phenomenon. With that, we model the dynamics of the objects using an ordinary differential equation (ODE). Our objective is now to estimate the unknown physical parameters, as well as the initial conditions, of this ODE. Hence, we additionally learn a video generation model that is able to render a video that depicts objects which follow a specific physical model depending on respective physical parameters. For estimating these physical parameters directly from an input video, we utilise a photometric loss that imposes that the generated video is similar to the input video.

## 3.1 Modeling the dynamics

For most of the dynamics that can be observed in nature, the temporal evolution of the state can be described by an ODE. For example, for a pendulum the state variables are the angle of deflection and the angular velocity, and a two dimensional first-order ODE can be used to describe the dynamics.

In general, we write $\dot{\mathbf{z}} = f(\mathbf{z}, t; \theta)$ to describe the ODE[1], where $\mathbf{z} \in \mathbb{R}^n$ denotes the state variable, $t \in \mathbb{R}$ denotes time and $\theta \in \mathbb{R}^m$ are the unknown physical parameters. Using the initial conditions $\mathbf{z}_0 \in \mathbb{R}^n$ at the initial time $t_0$, we can write the solution of the ODE as

$$\mathbf{z}(t; \mathbf{z}_0, \theta) = \mathbf{z}_0 + \int_{t_0}^{t} f(\mathbf{z}(\tau), \tau; \theta) \, \mathrm{d}\tau. \tag{1}$$

Note that the solution curve $\mathbf{z}(t; \mathbf{z}_0, \theta) \subset \mathbb{R}^n$ depends both on the unknown initial conditions $\mathbf{z}_0$, as well as on the unknown physical parameters $\theta$.

In practice, the solution to Eq. (1) is typically approximated by numeric integration. In our context of physical parameter estimation from videos, we build upon the recent work by Chen et al. (2018), who proposed an approach to compute gradients of the solution curve of an ODE with respect to its parameters. With that, it becomes possible to differentiate through the solution in Eq. (1) and therefore we can use gradient-based methods to estimate $\mathbf{z}_0$ and $\theta$.

## 3.2 Differentiable rendering of the video frames

To render the video frames, we draw inspiration from the recent advances in neural implicit representations. To this end, we use a static representation to model the background, which we combine with a appearance and shape representation of dynamic foreground objects. By composing the learned background with the dynamic foreground objects, whose poses are determined by the solution of the ODE encoding the physical phenomenon, we obtain a dynamic representation of the overall scene. Doing so allows us to query the color values on a pixel grid, so that we are able to render video frames in a differentiable manner. Fig. 2 shows an overview of the approach.

**Representation of background.** The static background is modeled by a function $F(\cdot; \theta_{\text{bg}})$ that maps a 2D location $\mathbf{x}$ to an appearance value $\mathbf{c} \in \mathbb{R}^C$, where $C$ denotes the number of appearance channels (e.g. RGB colors). The function $F(\cdot; \theta_{\text{bg}})$ encodes the appearance of the background and is represented as a neural network with learnable parameters $\theta_{\text{bg}}$. To improve the ability of the neural network to learn high frequency variations in appearance, we use Fourier features (Tancik et al., 2020), so that the input location $\mathbf{x} \in \mathbb{R}^2$ is mapped to a higher-frequency vector $\gamma(\mathbf{x}) \in \mathbb{R}^{4N_{\text{Fourier}}+2}$, where $N_{\text{Fourier}}$ is the numbers of frequencies used. The full representation of the background then reads $c_{\text{bg}}(\mathbf{x}) = F(\gamma(\mathbf{x}); \theta_{\text{bg}})$. For a more detailed discussion of the architecture, we refer to App. A.

**Representation of dynamic objects.** To compose the static background and the dynamically moving objects into the full scene, we draw inspiration from Ost et al. (2021) who use implicit representations to represent color and shape in a scene graph to decompose a dynamic scene into a background representation and dynamically moving local representations. A drawback of their work is that they do not use a physical model to constrain the dynamics, and therefore strong supervisory signals like the trajectories and the dimensions of the bounding boxes are essential. In our case, each dynamic object is represented in terms of a local neural implicit representation, which is then placed in the overall scene based on the time-dependent spatial transformation $T_t = T(\mathbf{z}(t; \mathbf{z}_0, \theta_{\text{ode}}), \theta_+)$. This transformation is parameterized by the unknown initial condition $\mathbf{z}_0$, the physical parameters $\theta_{\text{ode}}$ of the ODE, and possibly additional parameters $\theta_+$. As such, these parameters determine the transformation from the global coordinate system of the background to the local coordinate system.

---

[1] W.l.o.g. we only consider first-order ODEs here, since it is always possible to reduce the order to one by introducing additional state variables.

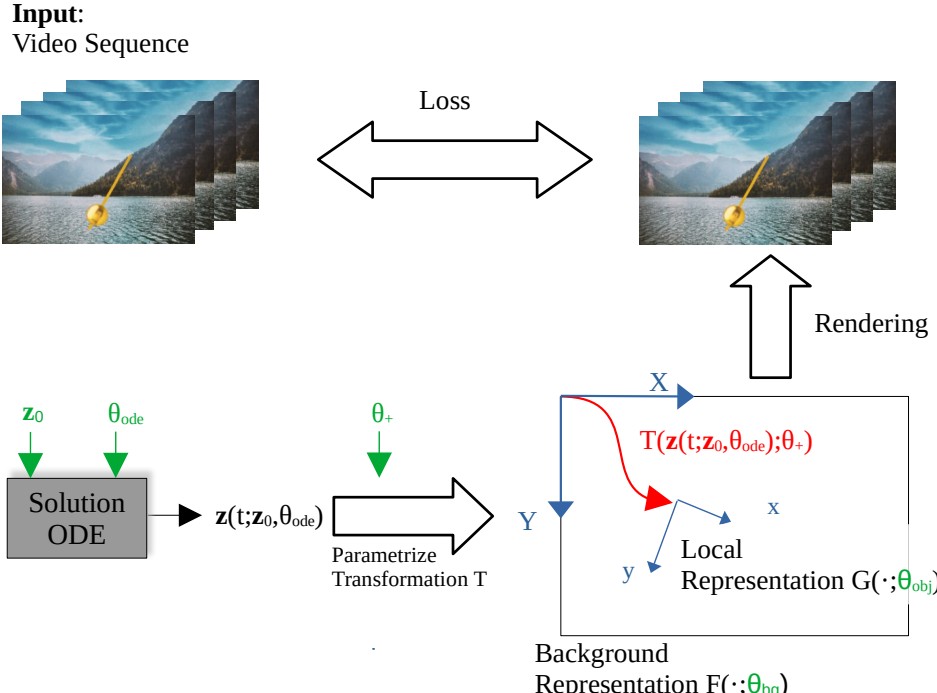

Figure 2: Overview of our approach. The dynamics in the video are modelled by an ordinary differential equation, which is solved depending on unknown initial conditions $\mathbf{z}_0$ and unknown physical parameters $\theta_{\mathrm{ode}}$. The solution curve $\mathbf{z}\left(t; \mathbf{z}_0, \theta_{\mathrm{ode}}\right)$ is used to parametrize a time-dependent transformation $T\left(\mathbf{z}\left(t; \mathbf{z}_0, \theta_{\mathrm{ode}}\right), \theta_+\right)$ from the global coordinates $XY$ of the background to the local coordinates $xy$ of the moving object. The functions $F(\cdot; \theta_{\mathrm{bg}})$ and $G(\cdot; \theta_{\mathrm{obj}})$ are neural networks that model the appearance of the background and of the object, respectively. We can estimate the unknown physical parameters for a given video based on a rendering loss which penalizes the discrepancy between the input video frames and the rendered video. All estimated parameters and network weights are shown in green in the figure.

Similarly as the background, the appearance of each individual dynamic object is modelled in terms of an implicit neural representation (in the local coordinate system). In contrast to the background, we augment the color output $c \in \mathbb{R}^C$ of the dynamic object representation with an additional opacity value $o \in [0, 1]$, which allows us to model objects with arbitrary shape. We write the representation of a dynamic object in the global coordinate system as $(c_{\mathrm{obj}}\left(\mathbf{x}\right), o\left(\boldsymbol{x}\right)) = G(\gamma\left(\boldsymbol{x}'\right); \theta_{\mathrm{obj}})$, where $G(\cdot; \theta_{\mathrm{obj}})$ is represented as a neural network with weights $\theta_{\mathrm{obj}}$, $\gamma$ denotes the mapping to Fourier features, and $\boldsymbol{x}' = T_t(\boldsymbol{x})$ is the local coordinate representation of the (global) 2D location $\boldsymbol{x}$.

**Differentiable rendering.** For rendering we evaluate the composed scene appearance at a regular pixel grid, where we use the opacity value of the local object representation to blend the color of the background and the dynamic objects. To obtain the final color, for all positions $\boldsymbol{x}$ of the pixel grid we evaluate the equation

$$c(\boldsymbol{x}, t) = (1 - o(\boldsymbol{x}))\, c_{\mathrm{bg}}(\boldsymbol{x}) + o(\boldsymbol{x}) c_{\mathrm{obj}}(\boldsymbol{x}). \tag{2}$$

Note that due to the time dependence of the transformation $T_t$, the color value for pixel $\boldsymbol{x}$ is also time dependent, which allows us to render the frames of the sequence over time.

### 3.3 Loss function

We jointly optimize for the parameters of the neural implicit representations $\theta_{\mathrm{bg}}$ and $\theta_{\mathrm{obj}}$ and estimate the physical parameters $\theta_{\mathrm{ode}}, \mathbf{z}_0$ and $\theta_+$ of the dynamics and the transformation. To this end, we use a simple photometric loss defined over all the pixel values, which reads

$$\mathcal{L} = \frac{1}{|\mathcal{I}|\,|\mathcal{T}|} \sum_{t \in \mathcal{T}} \sum_{x \in \mathcal{I}} d(I\left(\boldsymbol{x}, t\right), c\left(\boldsymbol{x}, t\right)), \tag{3}$$

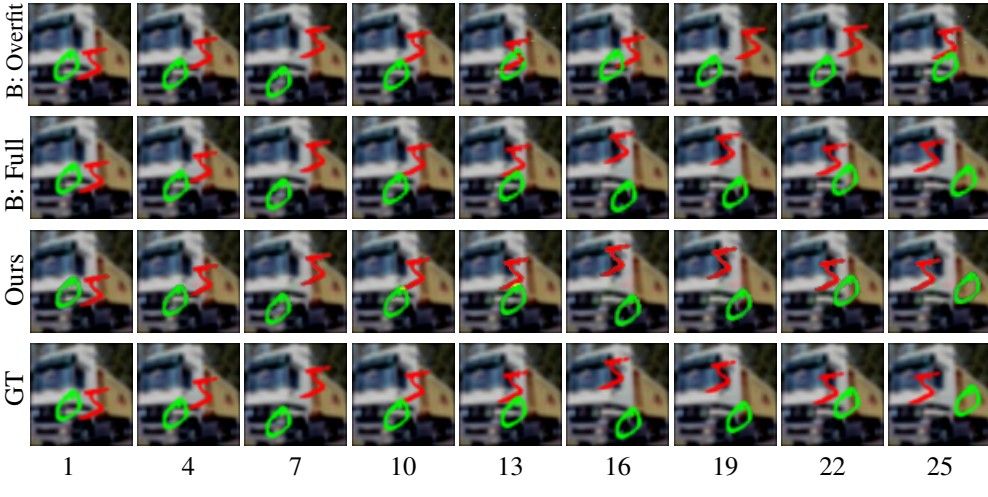

Figure 3: Two masses spring system in which MNIST digits are connected by an (invisible) spring. Reconstruction and prediction for test sequence 6. The arrow indicates the prediction start. For the spring constant and equilibrium distance $(k, l)$ the different methods achieve the following relative errors: $(2.7\%, 81.0\%)$ (B: Overfit), $(3.7\%, 1.8\%)$ (B: Full), and $(\mathbf{0.6}\%, \mathbf{1.7}\%)$ (Ours).

where $d$ computes the discrepancy between its two inputs, $\mathcal{T}$ is the set of all given time steps, $\mathcal{I}$ is the set of all pixel coordinates at the current resolution (see next section) and $I(\boldsymbol{x}, t)$ are the given images. To capture information on multiple scales we employ an image pyramid scheme. More details can be found in App. C.

## 4 EXPERIMENTS

We use two challenging physical models to experimentally evaluate our proposed approach. To analyze our method and to compare to previous work, we first consider synthetically created data. Afterwards, we show that our method achives promising results also on real-world data. For details about the ODEs describing the dynamics, additional implementation details, an ablation study, as well as additional results we refer the reader to the Appendix.

Although several learning-based approaches that infer physical models from image data have been proposed (de Avila Belbute-Peres et al., 2018; Jaques et al., 2020; Kandukuri et al., 2020; Zhong & Leonard, 2020; Toth et al., 2020), existing approaches are particularly tailored towards settings with large training corpora. However, these methods typically suffer from a decreasing estimation accuracy in scarce training data regimes, or if out of distribution generalization is required (cf. App. F). In contrast, our proposed approach is able to predict physical parameters from a single short video clip. Due to the lack of existing baselines tailored towards estimation from a single video, we adapt the recent work of Jaques et al. (2020) and Zhong & Leonard (2020) to act as baseline methods.

### 4.1 TWO MASSES SPRING SYSTEM

We consider the example of two moving MNIST digits connected by an (invisible) spring on a CIFAR background, in a similar spirit to Jaques et al. (2020), see Fig. 3. Besides the initial positions and velocities, the spring constant $k$ and the equilibrium distance $l$ of the connecting spring need to be identified for the dynamics model. For a more detailed description of the model see App. B.1.

The approach of Jaques et al. (2020) uses a learnable encoder and velocity estimator to obtain positions and initial velocities of a known number of objects from the video frames. After integrating the known parametric model, they use a learnable coordinate-consistent decoder in combination with learned object masks and colors to render frames from the integrated trajectories. Using a photometric loss they require 5000 sequences of different runs of the same two masses spring system to train the model and identify the parameters. In order to compare their method to our work in the set-

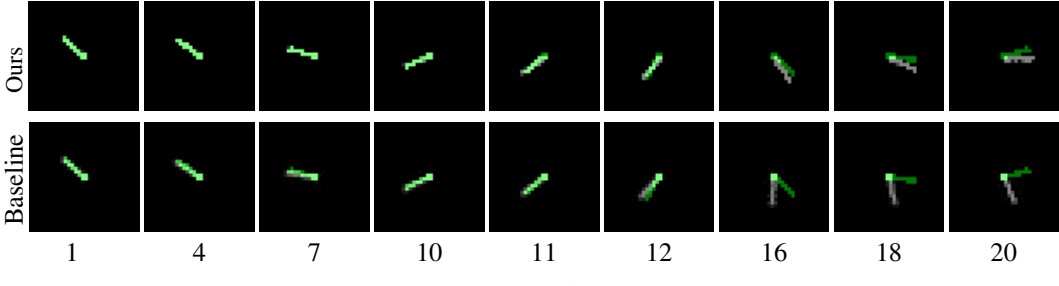

Figure 4: Prediction when training with the first $N = 10$ frames of sequence 1. Each image shows the prediction of the respective method in white, and the ground truth as green overlay. For both methods, the prediction of images seen during training (frames 1,4,7,10) works well. For unseen data (frames 11,12,16,18,20), our method (*top*) leads to more reliable predictions, meaning that our physical parameter estimation is more accurate.

ting of parameter estimation from single video, in addition to their model trained on the full dataset ('B: Full'), we also consider their model trained for an individual sequences of the test dataset ('B: Overfit').

We fit our model to sequences from the test dataset, where we use two local representations and parametrize the spatial transformation as shown in App. B.1. By using the maximum of both masks as foreground mask, we enable the model to identify the object layering. We find that for training it is necessary to gradually build up the sequence of frames over training. We start with only two frames and add the respective next frame after 60 epochs. Also, the model appears to have a scale freedom in terms of the equilibrium length and the points where the spring is attached to the digits.[2] We therefore add an additional loss to keep the spring attachment close to the center of the bounding box of the digits in the first frame. We observe similar effects when overfitting the model of Jaques et al. (2020) to a single sequence. When training on the full dataset, the effect seems to be averaged out and is not observed.

Fig. 3 shows a qualitative comparison of our results to the baseline of Jaques et al. (2020), where the latter is trained in the two settings explained above. We observe, that for this sequence all approaches yield reasonable results for the reconstruction of the training frames. However, for prediction the overfitted model of Jaques et al. (2020) performs significantly worse, indicating that the physical model is poorly identified from a single video. The baseline trained on the full dataset yields results that are slightly worse than our results. We see that in both cases the parameters are identified correctly. The fact that we achieve comparable results while using significantly less data highlights the advantage of combining the explicit dynamics model with the implicit representation for the objects. Note that we chose sequence 6 since it yielded the best results for the baseline. More results can be found in app. E.1.

## 4.2 NONLINEAR DAMPED PENDULUM

We use synthetically created videos of a nonlinear damped pendulum to compare our method to the previous work of Zhong & Leonard (2020) and also to show the ability of our approach to handle high resolution videos. The equations describing the pendulum dynamics can be found in App. B.2.

**Comparison to Lagrangian Variational Autoencoder.** We use the dataset of Zhong & Leonard (2020) containing several sequences of a simple pendulum (each comprising 20 frames), which was created by the OpenAI Gym simulator (Brockman et al., 2016). The method by Zhong & Leonard (2020) uses a coordinate aware encoder to obtain the distribution of the initial state from object masks. After sampling, the initial state is integrated using a learnable Lagrangian function parametrizing the dynamics of the system and a coordinate aware decoder is used to render frames from the trajectories. We train the model using only the first $N$ frames of a single sequence as the training data (with no external control input), effectively overfitting the model to each sequence.

---

[2]Intuitively, if the motion is only in one direction we can vary the equilibrium length and adjust the spring attachments without changing the observed motion. Similar effects are present in a 2D motion.

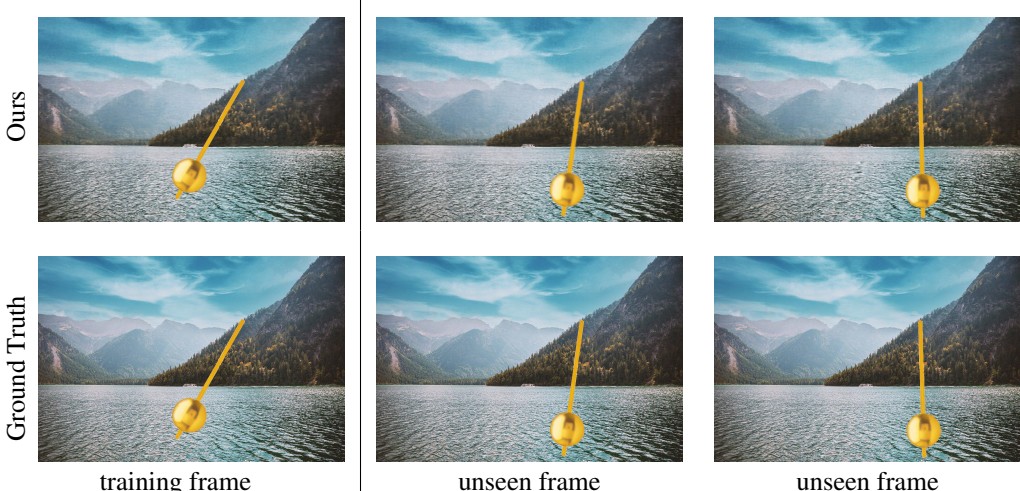

Figure 5: Rendered frames for the high-resolution ($1280 \times 854$ pixel) lake sequence when training on the first 10 frames out of a total of 25 frames. While the first image is part of the training set, the two remaining images are frame 20 and 25 of the full sequence and have not been seen during training. We see that our method produces realistic reconstructions of the scene even for physical states that are not seen during training. Best viewed on screen with magnification.

Similar to the baseline, we assume no damping and a known pivot point $A$ in the middle of the frame to train our model. Since this dataset does not include image data, we only use the loss on the object mask, and train our model in this modified setup using the same frames as for the baseline.

To evaluate the performance of each method in identifying the underlying dynamics, we compare the prediction of the unseen frames of the same sequence. Qualitative results are presented in Fig. 4. We can observe that both methods fit the given training data very well, however, in the baseline the pendulum motion significantly slows down for unseen time steps and thus it is unable to obtain accurate predictions for unseen data. We emphasize that this happens because the method requires significantly larger training datasets, so that it performs poorly in the single-video setting considered in this paper. In contrast, our method shows a significantly better performance, which highlights the strength of directly modelling physical phenomena to constrain the learnable dynamics in an analysis-by-synthesis manner. Due to aliasing effects that arise from the low resolution of the frames, our method does not give perfect predictions, however, if we use high-resolution images for our method we achieve nearly perfect reconstruction as we show in Fig. 5 and Fig. 6. For a quantitative comparison and further experimental details see App. E.2.

**High resolution videos.** In contrast to the baseline, our approach is able to handle high-resolution videos with complex background and pendulum shapes and textures. In this case, our approach accurately the parameters for the full pendulum model as we show in Fig. 5.

For this experiment we created several videos by simulating a pendulum with known parameters and then rendering the pendulum on top of an image. Qualitative results of fitting our model to the lake scene can be seen in Fig. 5. We see that our model produces photorealistic renderings of the scene, even for the predicted frames. The renderings of other scenes are shown in App. E. As we show in Fig. 6 and Table 1, it is necessary that the frames in the training set cover a sufficient portion of the motion to enable a correct estimation of the physical parameters.

### 4.3 REAL PENDULUM VIDEO

We now show that our approach is even able to infer physical parameters from real world data. We recorded the pendulum motion shown in Fig. 1. The pendulum is mounted almost frictionless and due to its high weight we do not expect large air drag effects either. The video was recorded with a smartphone, which leads to noticeable real-world noise such as motion blur, however, the proposed method still manages to produce convincing results. The pseudo groundtruth segmentation masks are generated semi-manually by using GrabCut (Rother et al., 2004) and exhibit significant noise

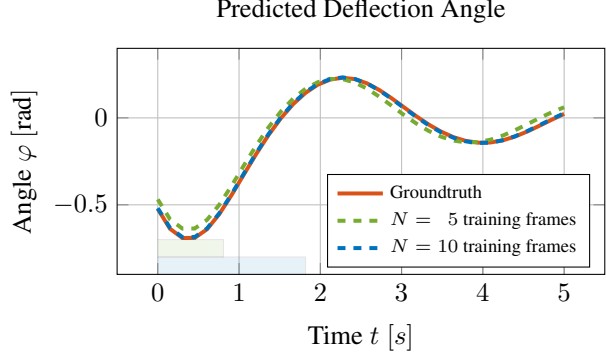
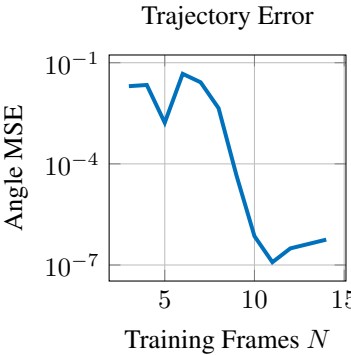

Figure 6: Prediction evaluation on the lake scene. The estimated deflection angle $\varphi$ is close to the groundtruth for $N = 5$ training frames and virtually identical for $N = 10$ training frames even during extrapolation (*left*). We show the time span covered by training images with transparent boxes above the x axis. The trajectory error, measured by the MSE between the groundtruth deflection angle and predicted deflection angle for $t \in [0, 5]$, improves with the number of training frames $N$ (*right*). This shows that a sufficient number of frames is necessary to constrain the model, which can then predict a nearly perfect trajectory.

that is also handled well by the proposed model. We extract every third frame from the video s.t. there are 10 extracted frames per second and use the first 10 frames for training.

We use the the full damped pendulum model to estimate the physical parameters of the pendulum motion. The damping is estimated as $c = 4.7 \cdot 10^{-13}$, which matches our expectation for this low friction setting. For the pendulum length we note from Eq. (6) that the estimated length $l = 27.7\,\mathrm{cm}$ is a real world quantity without scale ambiguity. Therefore, we can compare it to $l_{\mathrm{measured}} = 27.1\,\mathrm{cm}$ which we obtained by measuring the length from the pivot point to the estimated center of gravity of the pendulum using a ruler. We would like to emphazise, that the very good correspondence shows, that we are able to estimate scale in a monocular video from a pendulum motion.

## 5 CONCLUSION

In this work we presented a solution for learning a physical model from an image sequence that depicts some physical phenomenon. To this end, we proposed to combine neural implicit representations and neural ordinary differential equations in an analysis-by-synthesis fashion. Unlike existing learning-based approaches that require large training corpora, a single short video clip is sufficient for our approach. In contrast to prior works that use encoder-decoder architectures specifically tailored to 2D images, we built upon neural implicit representations that have been shown to give impressive results for 3D scene reconstruction. Therefore, the extension of the proposed method to 3D is a promising direction for future work.

We present diverse experiments in which the ODE parametrizes a rigid-body transformation between the background and the foreground objects, such as the pendulum motion. We emphasize that conceptually our model is not limited to rigid-body motions, and that it can directly be extended to other cases, for example to nonlinear transformations for modelling soft-body dynamics. The focus of this work is on learning a physical model of a phenomenon from a short video. Yet, the high fidelity of our model's renderings, together with the easy modifiability of the physical parameters, enables various computer graphics applications such as the artistic re-rendering of scenes, which we briefly demonstrate in the supplementary video. Overall, our per-scene model combines a unique set of favorable properties, including the interpretability of physical parameters, the ability to perform long-term predictions, and the synthesis of high-resolution images. We believe that our work may serve as inspiration for follow-up works on physics-based machine learning using neural implicit representations.

**Ethics statement.** This work attempts to learn interpretable physical models from video clips of physical phenomena. Our contribution is largely theoretical and we show experiments on synthetic data and limited real-world data. Nevertheless, as machine learning models achieve more human-like understanding of the real, physical world, it is paramount to ensure that they are deployed safely and according to strict ethical guidelines. While we think that the current state of our work will not disadvantage or advantage specific groups of people, we recommend a careful ethical evaluation of derivative works that aim to close the gap to human physical reasoning. A potential positive impact of this work is that it can be beneficial to people with lower financial resources, as it overcomes the need for expensive experimental gear to infer physical parameters, e.g. in the context of physics education.

**Reproducibility statement.** To ensure the reproducibility of this work we give architecture and training details in App. A and C. Furthermore, we will release our code upon acceptance, so that all experiments and figures shown in this paper can be reproduced.

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

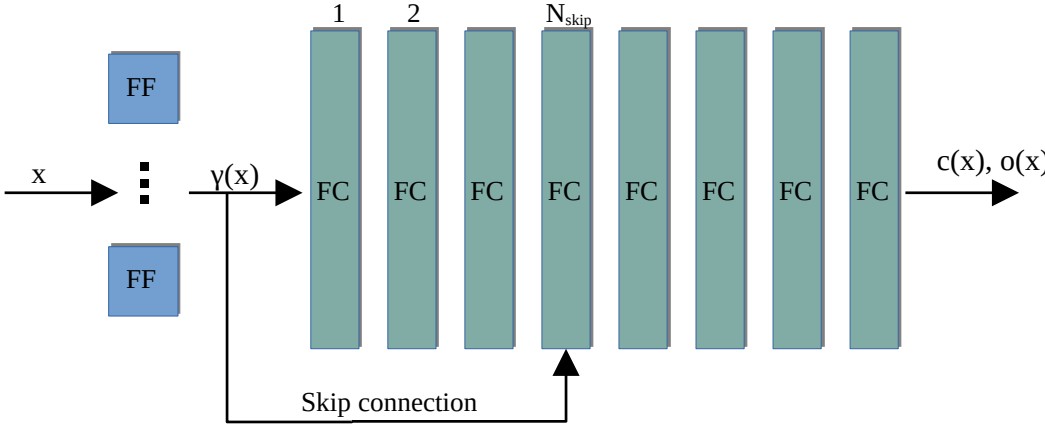

Figure 7: Overview of our architecture for the implicit shape and appearance representations. The input vector $\mathbf{x}$ is passed through a layer of $N_{\text{Fourier}}$ Fourier features (FF) to obtain the encoding $\gamma(\mathbf{x})$. The following neural network is constructed from $N_{\text{FC}}$ fully connected layers (FC) of width $W_{\text{FC}}$. We use ReLU activations between the layers. A skip connection is used to feed the encoding $\gamma(\mathbf{x})$ to the $N_{\text{skip}}$-th fully connected layer, where it is concatenated with the output of the previous layer. We feed the output of the last layer through a sigmoid function, to achieve values for the color $c$ and the opacity $o$ (only for the local representation) in the range $[0, 1]$.

## A  MODEL ARCHITECTURE

We adopt the architecture used in Mildenhall et al. (2020) for the implicit representations, see Fig. 7 for the basic structure. For the Fourier features we use a logarithmic scaling. The $i$-th of the $N_{\text{Fourier}}$ Fourier features is obtained as

$$\gamma_i(\mathbf{x}) = (\sin(2^i \mathbf{x}), \cos(2^i \mathbf{x})) \quad i = 0, \ldots N_{\text{Fourier}} - 1, \tag{4}$$

where $\sin(2^i \mathbf{x})$ for $\mathbf{x} \in \mathbb{R}^2$ means the element wise application of the sine function. We also include the original $\mathbf{x}$ in the encoding $\gamma(\mathbf{x})$.

## B  MODELS FOR THE DYNAMICS

### B.1  TWO MASSES SPRING SYSTEM

The system is modeled as two-body system where the dynamic of each object is described by Newton's second law of motion, i.e. $F = m\ddot{x}$, where $F$ is the force. Since only the ratio between force and mass can be identified without additional measurement, we fix $m = 1$, analogously to the work of Jaques et al. (2020). Using Hooke's law, we write the force applied to object $i$ by object $j$ as

$$F_{i,j} = -k \left( (p_i - p_j) - 2l \frac{p_i - p_j}{\|p_i - p_j\|} \right). \tag{5}$$

Using the position $p_i(t; k, l)$ of the objects to parametrize the trajectory of the local coordinate systems, we can write the time-dependent 2D spatial transformation to the local coordinate system $i$ as $T_t^{(i)}(x) = x - p_i(t; k, l)$, where $l$ and $k$ are learnable parameters.

### B.2  NONLINEAR DAMPED PENDULUM

A pendulum that is damped by air drag can be modelled as

$$\begin{bmatrix} \dot{\varphi} \\ \dot{\omega} \end{bmatrix} = \begin{bmatrix} \omega \\ -\frac{g}{l} \sin(\varphi) - c\omega |\omega| \end{bmatrix}, \tag{6}$$

where $\varphi \in \mathbb{R}$ is the deflection angle, $\omega \in \mathbb{R}$ is the angular velocity, $g$ is the (known) gravitational acceleration, $l > 0$ is the (physical) length of the pendulum, and $c > 0$ is the damping constant.

We use the solution curve $\varphi(t; l, c)$ to parameterize the time-dependent 2D spatial transformation as $T_t(x) = R(\varphi(t; l, c)) x + A$, where $R \in \mathrm{SO}(2)$ is a rotation matrix and $A \in \mathbb{R}^2$ is the pivot point of the pendulum. For the full model, the parameters $l$ and $c$ are learnable. For the sake of simplicity we assume that the gravitational acceleration $g$ always points downwards in the global image coordinate system.

## C   TRAINING DETAILS

In the following we provide additional training details.

### C.1   DISCREPANCY MEASURE FOR THE LOSS TERM

Unless stated otherwise, for our experiments we use $C = 4$ image channels, where the three first channels correspond to the RGB channels, and the last channel represents a mask of the foreground object. For the real world data, we obtained the objects masks using a semi-manual approach as described in Sec. 4.3. For the experiments on the synthetically created high resolution videos we used the masks constructed for the video creation for the experiments directly.

For the first three channels we define the discrepancy measure in terms of the mean square error as

$$d_{\mathrm{rgb}}(x, y) = \|x - y\|^2, \tag{7}$$

and for the mask in the last channel we consider the binary cross entropy loss, i.e.

$$d_{\mathrm{seg}}(x, y) = [x \log(y) + (1 - x) \log(1 - y)]. \tag{8}$$

With that, the overall discrepancy measure is given as

$$d(x, y) = d_{\mathrm{rgb}}(x_{1:3}, y_{1:3}) + \lambda_{\mathrm{seg}} d_{\mathrm{seg}}(x_4, y_4). \tag{9}$$

### C.2   OPTIMIZATION

We train our model using the Adam optimizer (Kingma & Ba, 2015) with exponential learning rate decay, which reads

$$r(e) = r_0 \cdot \beta^{e/n_{\mathrm{decay}}} \tag{10}$$

where $r(e)$ is the learning rate depending on the epoch $e$, $r_0$ is the initial learning rate, $\beta$ is the decay rate and $n_{\mathrm{decay}}$ is the decay step size.

One important aspect of the training is to use different learning rates for the parameters $\theta_{\mathrm{bg}}$ and $\theta_{\mathrm{obj}}$ of the implicit representations on the one hand and the physical parameters $\theta_{\mathrm{ode}}$, $\mathbf{z}_0$ and $\theta_+$ on the other hand.

In order to estimate the initial parameters of the ODE and the transformation for the pendulum we employ a heuristic that uses the information contained in the mask. To obtain an initial estimate for the pivot point $A$ we average all masks and use the the pixel with the highest value. To obtain an estimate for the initial angle, we perform a principal component analysis (PCA) on the pixel locations covered by the mask and use the angle between the first component and the vertical direction. The velocity is always initialized as $0$. We initialize the damping as $c = 1$ and the pendulum length as $l = 2\,\mathrm{m}$ for the synthetic experiments and $l = 0.4\,\mathrm{m}$ for the real world experiment.

### C.3   IMAGE PYRAMID

To capture information on multiple scales we employ an image pyramid scheme. Due to memory limitations, for large images we cannot evaluate all pixel values in one batch, and thus the classical approach that considers all stages of the image pyramid at once is not feasible in our setting. Therefore, during training, we sequentially traverse the image pyramid from the low-resolution levels towards the original high-resolution level. The idea is that the low resolution stages reveal global information about the movement of the object, whereas the later high-resolution stages allow to use finer details that improve the coarse estimates from the previous stages. To this end, we use a Binomial kernel of size $5 \times 5$ with stride two, which we repeatedly apply $N_{\mathrm{pyr}}$ times to reduce the original resolution of the image. We start the training using the coarsest level, and then switch to the next finer level every $n_{\mathrm{pyr}}$ steps.

| | | #Frames | $A$ | $c$ | $l$ | $x_0$ | Visual reconstruction |
|---|---|---|---|---|---|---|---|
| Lake | Full Loss | 5 | 1.06e-03 | 1.60e-03 | 7.23e-02 | 4.87e-02 | ✓ |
| | Full Loss | 10 | 1.71e-03 | **5.54e-04** | **1.07e-03** | **2.12e-03** | ✓ |
| | Only $d_{\mathrm{rgb}}$ | 10 | 4.82e-03 | 3.35e-02 | 9.65e-03 | 1.71e-02 | ✓ |
| | Only $d_{\mathrm{seg}}$ | 10 | **3.11e-04** | 4.12e-03 | 1.85e-03 | 2.17e-03 | ✗ |
| City | Full Loss | 5 | 5.87e-01 | 1.00e+00 | 1.81e-01 | 6.94e-01 | ✓ |
| | Full Loss | 10 | 1.90e-03 | **3.99e-03** | 4.73e-03 | 1.09e-02 | ✓ |
| | Only $d_{\mathrm{rgb}}$ | 10 | 9.20e-04 | 1.49e-02 | **5.70e-04** | **4.18e-03** | ✓ |
| | Only $d_{\mathrm{seg}}$ | 10 | **5.71e-04** | 9.00e-03 | 2.86e-03 | 7.44e-03 | ✗ |

Table 1: Relative error of the estimated parameters for the lake and city sequences. With $N = 10$ frames our model is able to accurately infer the physical parameters of the underlying physical phenomenon, while $N = 5$ frames are not sufficient to constrain all parameters. While the color as well as the segmentation loss alone give similar results to the combined loss, both losses are needed for the full model. The color loss enables the proposed approach to generate photorealistic renderings on top of physical parameter inference. While using only the color loss gives good accuracy for synthetic data, we observed that the segmentation loss, especially in combination with using image pyramids, greatly improves the convergence behaviour for initial conditions that are farther away from the groundtruth parameters. Overall, both losses are necessary to obtain accurate physical parameter estimates and photorealistic renderings in a robust manner.

## D  ABLATION STUDY

To motivate the chosen loss functions, we report the results for the parameter estimation with different loss function configurations in Table 1. Beyond the influence on the quality of the parameter estimation, another motivation to use the color loss is that it enables to learn the representation of the appearance of the background and the object in the implicit representation. This allows for photo-realistic rendering of unseen predictions, as well as the re-rendering of scenes with modified physical parameters, effectively allowing physical scene editing. For the mask loss, on the other hand, we have found that it makes the estimation process more robust to suboptimal initializations of the physical parameters.

## E  FURTHER EXPERIMENTAL DETAILS AND RESULTS

In the following we consider specific details for the different experiments.

### E.1  TWO MASSES SPRING SYSTEM

**Experimental details.**     We use both loss terms and set $\lambda_{\mathrm{seg}} = 0.01$ to balance them. Additionally, we use an MSE loss to keep the center of the bounding boxes of the digits close to the origin of the local representations in the first frame. This fixes the scale problem related to the equilibrium length described in the main text. Moreover, we use another MSE loss term to keep the opacity value close to zero outside of (but close to) the visible area. We found this to be necessary, since otherwise artefacts might appear in the extrapolation when previously unseen parts of the mask appear in the visible area.

For the background we use an implicit representation with $N_{\mathrm{Fourier}} = 6$ Fourier features, $N_{\mathrm{FC}} = 8$ fully connected layers of width $W_{\mathrm{FC}} = 128$ and an input skip to layer number $N_{\mathrm{skip}} = 4$. For the local object representation we use $N_{\mathrm{Fourier}} = 8$ Fourier features, $N_{\mathrm{FC}} = 8$ fully connected layers of width $W_{\mathrm{FC}} = 128$ and an input skip to layer number $N_{\mathrm{skip}} = 4$.

We use an initial learning rate of $r_{\mathrm{MLP, 0}} = 0.001$ for the parameters of the implicit representations and $r_{\mathrm{param, 0}} = 0.01$ for the physical parameters. We set $\beta_{\mathrm{MLP}} = 0.99954, n_{\mathrm{decay,MLP}} = 50, \beta_{\mathrm{param}} = 0.95$ and $n_{\mathrm{decay,param}} = 100$.

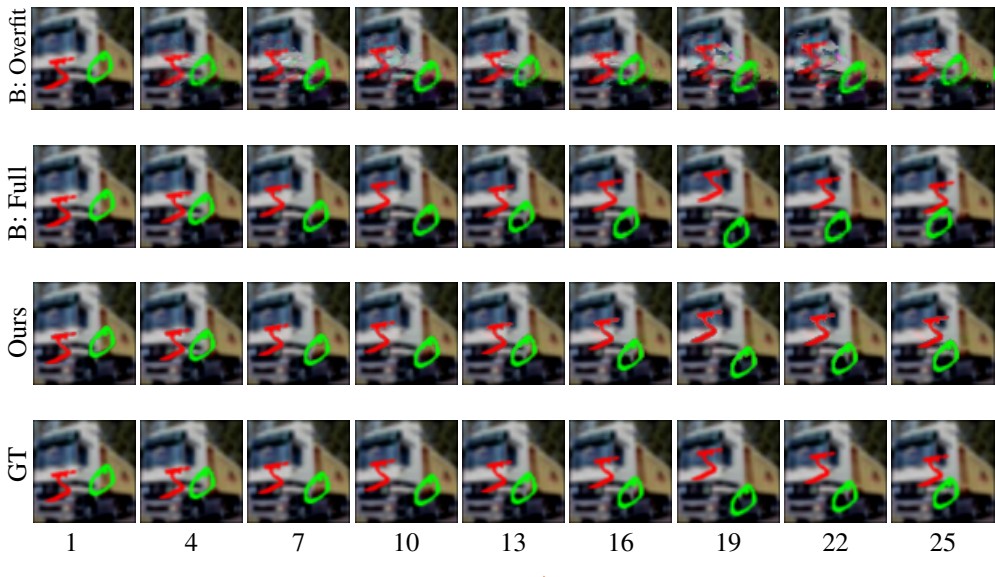

Figure 8: Two masses spring system, where MNIST digits are connected by an (invisible) spring. Reconstruction and prediction for test sequence 0. The arrow indicates where the prediciton starts. For the spring constant and equilibrium distance $(k, l)$ the different methods achieve the following relative errors respectively: $(20.4\%, 57.6\%)$ (B: Overfit), $(3.7\%, \mathbf{1.8}\%)$ (B: Full), and $(\mathbf{0.35}\%, 6.7\%)$ (Ours).

For the image pyramid we use $N_{\text{pyr}} = 1$ stage and step up the pyramid every $n_{\text{pyr}} = 200$ epochs. We train for 1500 epochs, where one epoch is completed, when all the pixels in the current resolution have been considered.

**Additional results.** In Fig. 8 and Fig. 9 we present additional results for sequence 0 and sequence 1 of the test dataset. We see, that for both sequences, overfitting the baseline is not able to produce a reasonable extrapolation of the data and even produces artifacts for the reconstruction part of the sequence. One reason for this is that the model is unable to identify the physical parameters correctly as can be seen by the large relative errors. Our model, on the other hand, is able to estimate the parameters with high accuracy that is even slightly better than the baseline trained on the full training dataset, which again shows the strength of our approach, considering, that we use a single video as input.

## E.2 COMPARISON WITH THE LAGRANGIAN VARIATIONAL AUTOENCODER

**Experimental details.** The data used in this experiment does not include image data, therefore we do not use $d_{\text{rgb}}$ and set $\lambda_{\text{seg}} = 1$. Since the predicted masks are obtained only from the local representation, we do not use an implicit representation for the background in this example. For the local representation we use $N_{\text{Fourier}} = 4$ Fourier features, $N_{\text{FC}} = 6$ fully connected layers of width $W_{\text{FC}} = 64$ and an input skip to layer number $N_{\text{skip}} = 3$.

We use an initial learning rate of $r_{\text{MLP}, 0} = 0.001$ for the parameters of the implicit representations and $r_{\text{param}, 0} = 0.01$ for the physical parameters. We set $\beta_{\text{MLP}} = 0.9954, n_{\text{decay,MLP}} = 10, \beta_{\text{param}} = 0.995$ and $n_{\text{decay,param}} = 50$.

For the image pyramid we use $N_{\text{pyr}} = 2$ stages and step up the pyramid every $n_{\text{pyr}} = 75$ epochs. We train for 1500 epochs, where one epoch is completed, when all the pixels in the current resolution have been considered.

**Quantitative comparison.** To quantitatively compare the temporal prediction ability of our approach with the baseline, we follow the procedure by Zhong & Leonard (2020) and report the average mean squared error (MSE) between the predicted and the ground truth mask for the frames

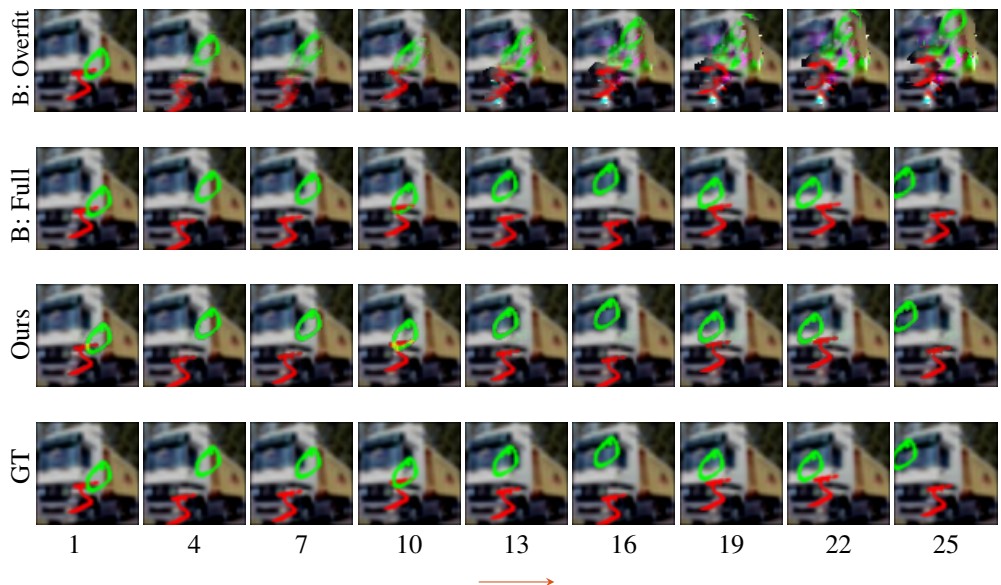

Figure 9: Two masses spring system, where MNIST digits are connected by an (invisible) spring. Reconstruction and prediction for test sequence 1. The arrow indicates where the prediciton starts. For the spring constant and equilibrium distance $(k, l)$ the different methods achieve the following relative errors respectively: $(13.6\%, 90.9\%)$ (B: Overfit), $(3.7\%, 1.8\%)$ (B: Full), and $(\mathbf{0.1\%}, \mathbf{0.3\%})$ (Ours).

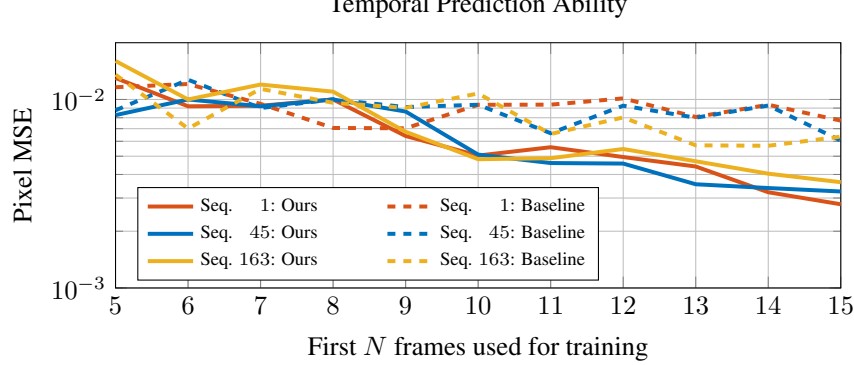

Figure 10: Temporal prediction ability of our approach and the approach of Zhong & Leonard (2020) overfitted to a single sequence (baseline).We report the average MSE (Pixel MSE) of the predicted masks for the entire sequence. The horizontal axis indicates the number of frames used for training, and the vertical axis shows the resulting error.

of the full sequence, which we denote as *pixel MSE* for consistency with the previous work. The results for randomly chosen sequences of the dataset are presented in Fig. 10. We can observe that the predictive power for both methods is limited when only a few frames are available to infer the underlying dynamics. However, with an increasing number of frames, our method becomes able to reconstruct the physics more consistently, while the baseline does not noticeably benefit from more training frames. We believe that this is because the baseline method overfits to the given frames, whereas our method infers actual physical parameters.

### E.3 EXPERIMENTS WITH HIGH RESOLUTION VIDEOS AND THE REAL WORLD VIDEO

**Experimental details.** We use the same architecture for the high resolution synthetic and the real-world video sequences. We use both loss terms and set $\lambda_{\text{seg}} = 0.03$ to balance them. For the background we use an implicit representation with $N_{\text{Fourier}} = 10$ Fourier features, $N_{\text{FC}} = 8$ fully connected layers of width $W_{\text{FC}} = 128$ and an input skip to layer number $N_{\text{skip}} = 4$. For the local object representation we use $N_{\text{Fourier}} = 8$ Fourier features, $N_{\text{FC}} = 8$ fully connected layers of width $W_{\text{FC}} = 128$ and an input skip to layer number $N_{\text{skip}} = 4$.

We use an initial learning rate of $r_{\text{MLP}, 0} = 0.001$ for the parameters of the implicit representations and $r_{\text{param}, 0} = 0.05$ for the physical parameters. We set $\beta_{\text{MLP}} = 0.99954, n_{\text{decay,MLP}} = 10, \beta_{\text{param}} = 0.95$ and $n_{\text{decay,param}} = 100$.

For the image pyramid we use $N_{\text{pyr}} = 5$ stages and step up the pyramid every $n_{\text{pyr}} = 200$ epochs. We train for 1500 epochs, where one epoch is completed, when all the pixels in the current resolution have been considered.

**Additional results.** In the following we present additional rendering results. Fig. 11 and Fig. 12 show additional reconstruction and prediction results for additional synthetic high resolution scenes. To create the synthetic scenes we took the background images from https://pixabay.com/photos/lake-mountains-nature-outdoors-6627781/ (Lake), https://pixabay.com/photos/city-street-architecture-business-4667143/ (City) and https://pixabay.com/photos/apples-fruits-ripe-red-apples-6073599/ (Apple). Fig. 13 allows for a more detailed comparison for the results of the real pendulum video. We show the images and masks used for training on the real pendulum video in Fig. 14. Please also see our supplementary video for additional results on this data.

## F GENERALIZATION OF THE LAGRANGIAN VARATIONAL AUTOENCODER

One drawback of learning-based approaches for visual estimation of physical models is the poor generalization to data that deviates from the training data distribution. We confirm this for the fully (pre-)trained model of Zhong & Leonard (2020). While the Pixel MSE averaged over the full test set is $1.83 \cdot 10^{-3}$, the error increases to $1.22 \cdot 10^{-2}$ when we shift the frames of the test data set by as much as 1 pixel in each direction. This corresponds to the case of input videos, where the pivot point of the pendulum is not in the center of the image, which is different from the training data. This effect is visualized in Fig. 15, which shows the output of the model for sequence 2 of the test data set with zero control input, both in the original version and in the shifted version. We observe that the small shift of only one pixel in each direction leads to results that are significantly off, and not even the first frame is predicted correctly. While Zhong & Leonard (2020) propose to use a coordinate-aware encoder based on spatial transformers, this introduces additional complexity to the model. In contrast, our approach does not suffer from such issues.

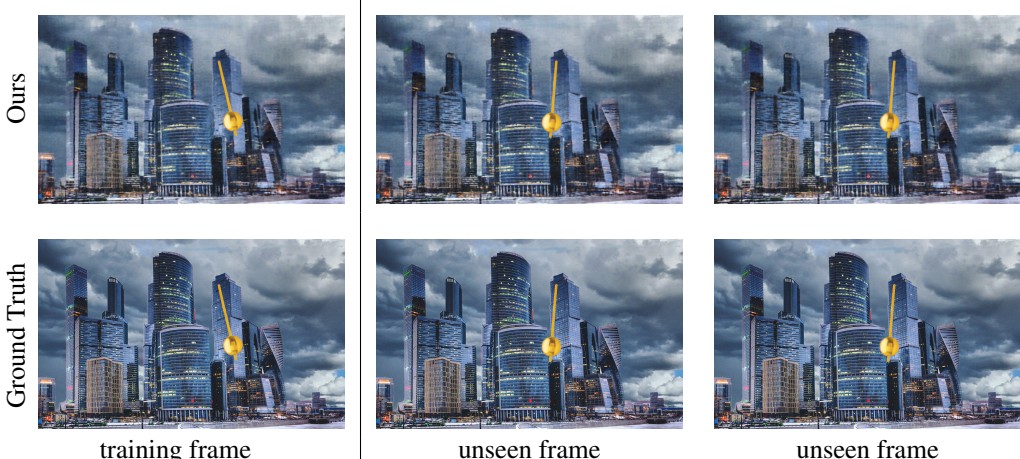

Figure 11: Reconstruction and prediction results for the city scene. The first frame is in the training set of 10 frames, while the two frames on the right are frame 20 and 25 of the sequence.

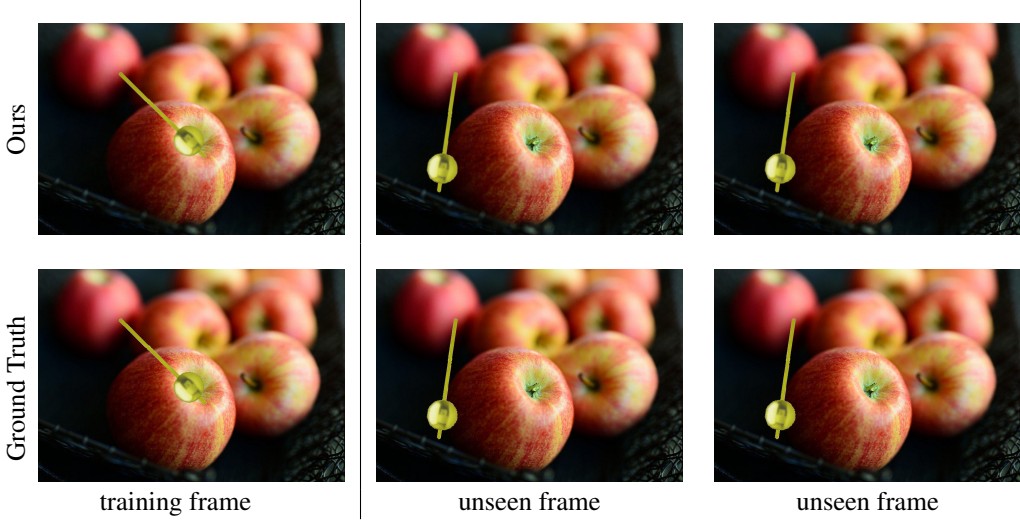

Figure 12: Reconstruction and prediction results for the apple scene. The first frame is in the training set of 10 frames, while the two frames on the right are frame 20 and 25 of the sequence.

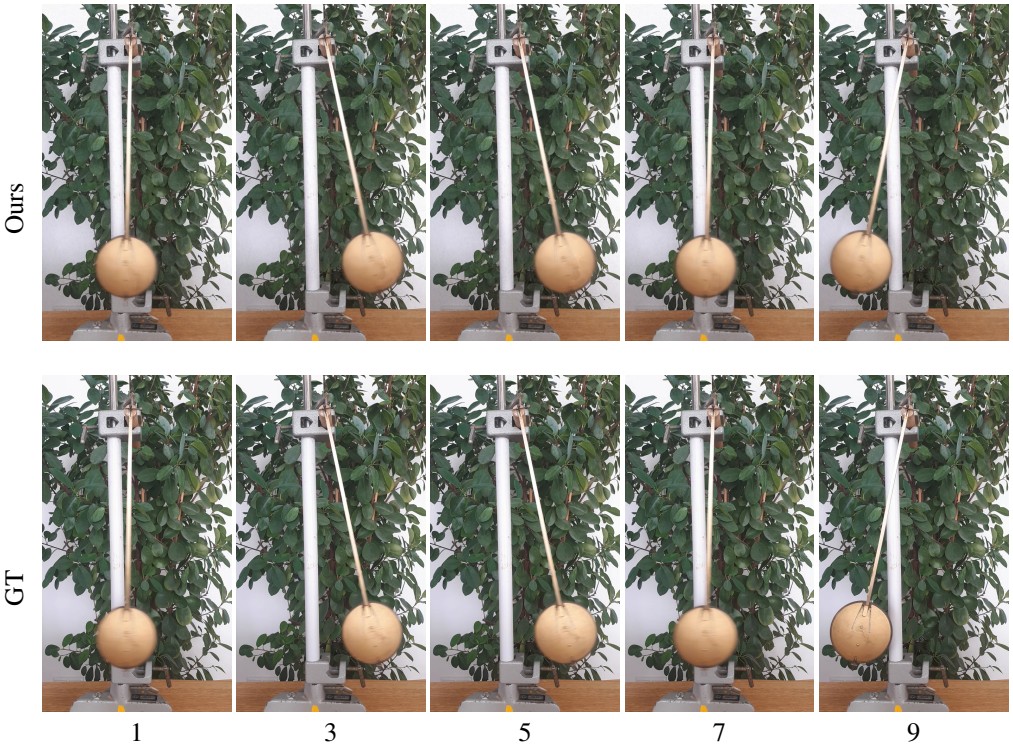

Figure 13: Comparison of the visual quality of the reconstruction for the real pendulum video trained on the sequence of 10 frames. The numbers indicate which frame of the sequence is shown. Best viewed on screen with magnification. Please see also our supplementary video.

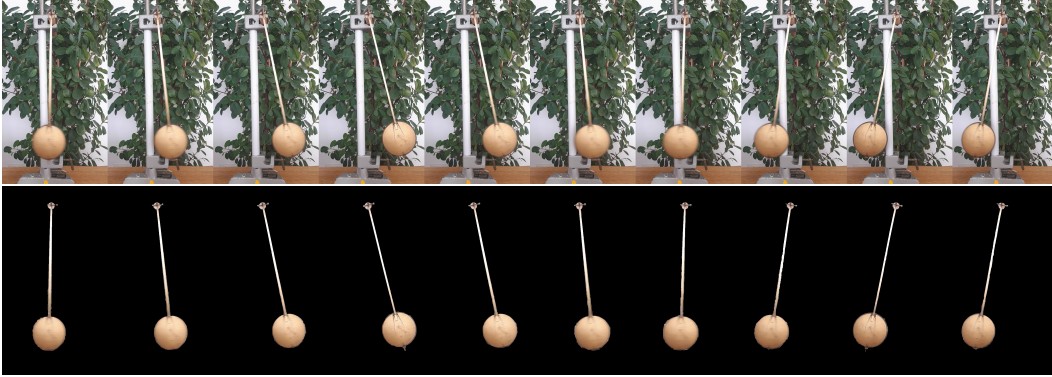

Figure 14: Training frames and segmentation masks for the real world video. Best viewed on a digital screen with magnification. Upon closer inspection the motion blur as well as segmentation mask error can be seen. The proposed approach can handle this real-world noise and produce compelling reconstructions and predictions shown in Fig. 13 and the supplementary video.

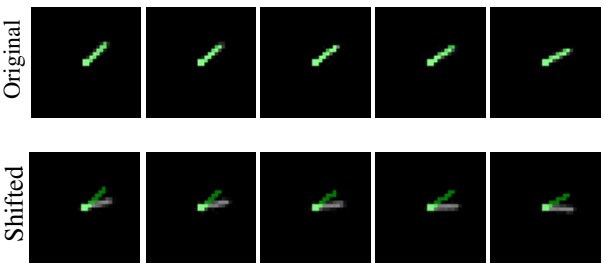

Figure 15: Prediction of the fully trained model of Zhong & Leonard (2020) for sequence 2 of the test dataset (with zero control). While the prediction for the original data is perfect, the prediction for shifting the frames by one pixel in each direction is significantly worse. This shows, that the model does not generalize well to input frames where the pivot point of the pendulum is not in the center of the frame.

