# OpenReview forum: "Neural Implicit Representations for Physical Parameter Inference from a Single Video"
_ICLR.cc/2022/Conference — ICLR 2022 Submitted_

### Official Review · Reviewer_fvJG · 2021-10-20

**Correctness:** 3
**Technical Novelty And Significance:** 3
**Empirical Novelty And Significance:** 3
**Recommendation:** 5
**Confidence:** 3

**Main Review:**

Strengths
1. The proposed approach is novel. Although there are methods that represents a dynamic object as a canonical object and object motions, this work innovatively models object motions as an ordinary differential equation, which can produce interpretable physical parameters.
2. The paper is well-written. I can understand the approach easily.
3. The technical details are complete.

Weaknesses
1. The validation experiments are not sufficient. The paper says that it can learn physical models from videos. However, there are only experiments on a nonlinear damped pendulum model. It is not sure that the proposed approach can work on other physical models.
2. The comparison experiments only compare the proposed approach with one baseline. I am not sure if there are other methods in the setting, but it would be better to construct some baselines to validate the effectiveness of the proposed components.
3. The ablation studies are not sufficient. There are some ablation studies that can be conducted, such as the iamge pyramid scheme, validating the decomposition of foreground and background, and the accuracies of the approach for captured objects in different motion states.

**Summary Of The Paper:**

Summary

This paper represents a video as a canonical implicit representation and a set of object motions that are described with an ordinary differential equation. It renders the representation into images and optimize its parameters by minimizing the difference between rendered images and target images. This work utilize an image pyramid for efficient training. It performs experimentes on a nonlinear damped pendulum in both synthetic and real-world settings.

Contributions
1. This paper represents nonlinear object motions with an ordinary differential equation, which enables them to recover interpretable physical parameters from videos.
2. It performs several experiments to validate the proposed approach.

**Summary Of The Review:**

The proposed approach is novel and interesting. But there need to be more experiments to validate the approach.

---

> ### Author Response · Authors · 2021-11-23
> **Response to Reviewer fvJG**
>
> We thank the reviewer for the insightful and constructive feedback, which will help us to further improve our work. We appreciate that the reviewer values the novelty of the paper and considers the paper to be well written and technically complete. In a response to all reviewers we have briefly listed the main changes to the manuscript, which are also highlighted in blue in the PDF. In the following, we will address the points raised in the review.
>
> &nbsp;
>
> **Experimental Evaluation and Baselines**
>
> We have included the example of two MNIST digits connected by a spring in the experimental section:
> 1. We show that our proposed method works for other physical systems, in this specific case two masses that are connected by a spring.
> 2. By comparing against the method proposed by Jaques et al. [1] we include another baseline as reference.
> 3. We show that the approach is able to handle two coupled dynamic objects.
> *[answer reproduced from our general response]*
>
> Overall, we believe that this experiment also helps in putting our work into the context of existing work.
>
> &nbsp;
>
> **Ablation Study**
>
> We thank the reviewer for proposing interesting further ablations. To quantify the convergence behaviour of the proposed model with and without the image pyramid, we would have to perform a large-scale experiment because the convergence depends on the seed. To gain statistical significance, many training are necessary and we thus had to prioritize the new experiments listed above over additional ablations.
>
> &nbsp;
>
> [1] Miguel Jaques, Michael Burke, and Timothy M. Hospedales.  Physics-as-inverse-graphics:  Unsupervised physical parameter estimation from video. ICLR 2020.

---

> > ### Comment · Reviewer_fvJG · 2021-11-29
> > **Response**
> >
> > I agree with other reviewers' concerns. So I decrease my score.

---

### Official Review · Reviewer_LTRa · 2021-11-01

**Correctness:** 3
**Technical Novelty And Significance:** 4
**Empirical Novelty And Significance:** 3
**Recommendation:** 6
**Confidence:** 2

**Main Review:**

I am not especially familiar with literature or work in physical parameter inference and the state-of-the-art and practices in the field. Thus my comments focus more on the usage of implicit neural representations, or coordinate-based networks, in this application.

In my opinion, the strengths of the paper are as follows:
1. It seems like the qualitative results significantly outperform the baseline method. Figure 3 shows that the proposed method generalizes to unseen frames significantly better than the baseline method. Figure 4 shows that this is especially the case when more training frames can be used to better infer physical parameters.
2. The paper is written very clearly (minus a few pieces of information I could not find, see weaknesses), and is overall enjoyable to read. I feel like I learned something about inferring physical parameters from only visual observations. I feel like this is an important problem, and especially could enable synthesis of realistic new visual content.

In my opinion, the weaknesses of the paper are as follows:
1. It seems like the results are very limited in terms of the types of situations they can be applied to. Only results with a pendulum overlaid over a synthetic background are shown, with one single real pendulum example? Is this formulation easy to extend to other physical models? If so, showing this would make the paper appear much stronger as this could be a more general tool for learning dynamics from only visual data.
2. Some clarity of explanation that seems to be missing
    - I don’t see any explanation of the “baseline method” which is compared to. I think much more detail needs to be given to this in the paper to understand if the method is providing a meaningful contribution or not. Similarly, why are no other inference methods that are mentioned in the related work compared to? It is hard to tell the magnitude of the contribution without this.
    - I think that the methods section on “representation of dynamic objects” could contain more information which makes it more clear how the method from Ost et al. (2021) is adapted to this application. What does the dynamic object coordinate-based network model, pixel colors on a coordinate domain? I understand they are warped in the overall scene conditioned on the physical parameters and initial conditions, but it’s not immediately clear how this is translated into an image which can then be composited on to the background.
3. I think it’s not immediately clear why a coordinate-based representation of the scene is really the useful tool here. Is it because rendering from the coordinate-based network is differentiable, allowing gradients to propagate back through the ODE solver and physical parameters? Because if not, I could see a potential baseline foregoing the neural representation of the video as a whole and optimizing physical parameters based on some deterministic rendering algorithm which warps pixels. For example, the background representation network seems unnecessary since it is completely static, why not just have a static image instead of a network to represent it? It would help the paper to have some explanation on this, and potentially ablate some of these contributions.


**Summary Of The Paper:**

The paper proposes a method which is capable of obtaining physical models of motion using only visual observations of the motion from a single video (no training data). This is done by combining a differentiable ODE solver with a coordinate-based network which reconstructs the video frames based on the ODE solution. Since this system is fully differentiable, the synthesized video loss can be used to infer the physical parameters and initial condition of the physical model along with the coordinate-based network parameters which represent the video. The work demonstrates that the physical parameters learned using this method are more accurate than those of baseline methods, and can be used to make long-term predictions of future frames of the video more accurately.

**Summary Of The Review:**

Overall, the paper describes a method which uses a coordinate-based network and ODE solver in conjunction to learn physical parameters (and their initial conditions) from a single video. This method is shown to outperform the “baseline” significantly, and seems to generate good results in terms of physical parameter estimation and video synthesis. Thus, I believe that this contribution is impactful, as I haven’t seen coordinate-based networks used in this way before. However, the amount of evaluation seems limited (only on the case of a pendulum), the comparisons to competing methods seem limited, and it’s not immediately clear to me why the coordinate-based network needs to be used for learning from a single video. With some clarification of this, I would be happy to raise my score, but based on my limited understanding I believe that the paper is borderline or slightly above.

**Update after Author Response**:
See response comment, I have chosen to retain my score.

---

> ### Author Response · Authors · 2021-11-23
> **Response to Reviewer LTRa**
>
> We thank the reviewer for the insightful and constructive feedback, which will help us to further improve our work. We are happy that the reviewer enjoyed reading our paper and shares our belief that estimating physical parameters from visual observations is an important topic. In a response to all reviewers, we have briefly listed the main changes to the manuscript, which are also highlighted in blue in the PDF. In the following, we will address the points raised in the review.
>
> &nbsp;
>
> **Experimental Evaluation**
>
> We have included the requested MNIST digits example from Jaques et al. in section 4.1. We believe that our work is now better put into the context of existing work.
> 1. We show that our proposed method works for other physical systems, in this specific case two masses that are connected by a spring.
> 2. By comparing against the method proposed by Jaques et al. [1] we include another baseline as reference.
> 3. We show that the approach is able to handle two coupled dynamic objects.
> *[answer reproduced from our general response]*
>
> &nbsp;
>
> **Clarification of Explanations**
>
> We appreciate the helpful feedback and have tried to strengthen the paper in regard to the raised points.
> * We have added a short summary of the methods of Zhong & Leonard and Jaques et al. which we added to the experimental section. We hope this helps to put our work better into context.
> * We have included more background to the approach of Ost et al. [2] to make it clearer how we adapt the ideas to our setting.
>
> &nbsp;
>
> **Use of Coordinate-based Representation**
>
> We have used neural coordinate-based representations as the backbone of our method because of their elegance, versatility, and accuracy. They have been shown to work remarkably well in 3D and produce near photorealistic results, which offers a clear direction for the generalization of the proposed method to 3D. We highly appreciate the proposed idea of a pixel-based baseline for representing the background. However, in this work we did not implement it since it would be hand-crafted specifically for the 2D scenario, thereby making the generalization to 3D more difficult. Nevertheless, this may be a promising direction for future work.
>
> &nbsp;
>
> [1] Jaques et al,  Physics-as-inverse-graphics: Unsupervised physical parameter estimation from video. ICLR 2020.
>
> [2] Ost et. al, Neural scene graphs for dynamic scenes. CVPR 2021.

---

> > ### Comment · Reviewer_LTRa · 2021-11-30
> > **Rebuttal Response**
> >
> > I appreciate the detailed rebuttal and changes to the manuscript. I will address my remaining concerns as follows:
> >
> > **Experimental Evaluation**:
> > I appreciate the updated results demonstrating the method in another physical system. However, I still am concerned that this example contains the same limitations as the original experiment, such as relying on separation into dynamic foreground and static background, and knowledge of the ground truth ODE equations. I understand that this can be viewed as a strength (a way to incorporate prior knowledge into the model), but it immediately limits the approach to scenarios where the dynamics are well-understood.
> >
> > **Use of Coordinate-based Representation**:
> > It makes sense that the use of a coordinate-based representation allows for the easier scaling to higher dimensions, but I think that for this argument to resonate much more strongly, then experiments outside of the 2D scenario need to be shown. Thus, I think that this paper would be significantly stronger and this argument would make significantly more sense of why a coordinate-based network is used, if there were experiments also capturing this 3D capability. Otherwise, in 2D only experiments, I think that a simple, hacky baseline should still be compared to.
> >
> > Based on my understanding of the paper, and the author response, I am not inclined to change my score any higher or lower. I think that the paper is interesting, and makes a contribution in the space of physical parameter estimation, but also would benefit from another round of peer review with more comprehensive experimental results.

---

### Official Review · Reviewer_SZKC · 2021-11-03

**Correctness:** 3
**Technical Novelty And Significance:** 2
**Empirical Novelty And Significance:** 2
**Recommendation:** 3
**Confidence:** 3

**Main Review:**

The paper fails to provide sufficient empirical support for the advantages of their approach. In particular, I feel like critical baselines are missing (e.g. Jaques et al. 2020) and pendulums are the only dynamics tested.

**Summary Of The Paper:**

This paper proposes a novel method for inferring a physical model from a single video. Their approach estimates the physical parameters and initial conditions of an ODE that describe the dynamics of the object, using neural implicit representations to render a video sequence based on the physical parameters. The authors perform experiments using simulate and real-world pendulum videos and compare to a Lagrangian VAE baseline.

**Summary Of The Review:**

As it stands currently, my feeling is that the evidence for the contribution and improvement of the proposed method over prior work is lacking.

---

> ### Author Response · Authors · 2021-11-23
> **Response to Reviewer SZKC**
>
> We thank the reviewer for the helpful feedback, which will help us to further improve our work. We appreciate that the reviewer values the novelty of the paper. In a response to all reviewers, we have briefly listed the main changes to the manuscript, which are also highlighted in blue in the PDF. In the following, we will address the points raised in the review.
>
> &nbsp;
>
> **Experimental Evaluation**
>
> We have included the requested MNIST digits example from Jaques et al. in section 4.1. We believe that our work is now better put into the context of existing work.
> 1. We show that our proposed method works for other physical systems, in this specific case two masses that are connected by a spring.
> 2. By comparing against the method proposed by Jaques et al. [1] we include another baseline as reference.
> 3. We show that the approach is able to handle two coupled dynamic objects.
> *[answer reproduced from our general response]*
>
> &nbsp;
>
> [1] Miguel Jaques, Michael Burke, and Timothy M. Hospedales.  Physics-as-inverse-graphics:  Unsupervised physical parameter estimation from video. ICLR 2020.

---

### Official Review · Reviewer_s2K9 · 2021-11-03

**Correctness:** 2
**Technical Novelty And Significance:** 2
**Empirical Novelty And Significance:** 1
**Recommendation:** 3
**Confidence:** 4

**Main Review:**

The paper studies the interesting problem of inferring the underlying physics of a scene given a video.

The paper is well written and the details clear to follow.

However, I have several different major concerns about the approach both in terms of novelty and evaluated empirical results, .

The underlying setting specified in this paper -- that of learning physics from a single video is more of a limitation in my opinion than a strength. To really learn the underlying physics of objects, more than a single video is needed. For example, if I were only to see a block fall once, there is only so much physics information I can gain. Instead, I think the much more interesting question is how we may learn physics of objects more generally from a dataset of different videos.

The underlying novelty of the project is also rather limited. There are variety of different works in NERF that have been fit to different dynamic scenes and video. This paper appears to build largely on top of [1], but swaps a neuralODE to represent the propogation of dynamics, which has also been done before [3].

The synthetic pendulum results seem very toy, with the pendulum rendered on a fake background. In general, the evaluation is the paper is insufficient -- only pendulum scenes are consider, and in that only a single video of the real world.

The underlying approach is also very domain specific. The underlying ground truth ODE equations for the pendulum are given.Furthermore, utilizing a NeuralODE seems limiting when modeling physical scenes. Many physical scenes do not conserve physics.

The paper does not show any downstream applications of the physics that is learned by the approach. Nor is the approach shown to generalize in any way to a new dataset

I would like to see a comparison where a non continuous neural network is used to predict the underlying dynamics of the approach.

The paper is also missing references several additional implicit representations for modeling dynamic scenes which are relevant [2-4]. [3] for example, also utilizes an NeuralODE to parameterize the underlying dynamics of a scene.

[1] Ost et. al, Neural scene graphs for dynamic scenes. CVPR 2021.
[2] Xian et. al, Space-time  Neural  Irradiance  Fields  for  Free-Viewpoint Video, CVPR 2021.
[3] Du et. al, Neural Radiance Flow for 4D View Synthesis and Video Processing, ICCV 2021.
[4] Li et. al, Neural Scene Flow Fields for Space-Time ViewSynthesis of Dynamic Scenes, CVPR 2021.

**Summary Of The Paper:**

This paper proposes to infer from a single video, the underlying physical parameters of a system. To do this, NeuralODE is combined with a neural radiance field to fit the desired video. The physical parameters learned by a NeuralODE may then be reutilized for simulation of the video under novel dynamics

**Summary Of The Review:**

The paper has a variety of issues which do not (in my opinion) make it fit for presentation at this time.

---

> ### Author Response · Authors · 2021-11-23
> **Response to Reviewer s2K9**
>
> We thank the reviewer for the important feedback, which will help us to further improve our work. We appreciate that the reviewer values the clear presentation of the paper and shares our interest in inferring the underlying physics of a scene from a given video. In a response to all reviewers, we have briefly listed the main changes to the manuscript, which are also highlighted in blue in the PDF. In the following, we will address the points raised in the review.
>
> &nbsp;
>
> **Physical Parameter Estimation from a Single Video**
>
> We would like to emphasize that we specifically target the setting of estimating physical parameters from a single video, which we consider a challenging scenario (in agreement with the reviewer's assessment). While generally more data would be desirable, we believe that there exist many scenarios where it is simply not possible to collect enough data, or where it is difficult to collect data that covers the underlying distribution sufficiently. Our approach uses the prior knowledge/assumption of the physical model to enable the estimation of physical parameters from a single video and therefore is tailored to the aforementioned settings in which large amounts of data are not available. We agree that the field of designing models that are able to infer more general physical models is very exciting, but judging from previous work and concurring to the reviewer's assessment, we believe that this requires large amounts of data, while our approach requires only a single video (but therefore requires detailed knowledge of the physical model).
>
> &nbsp;
>
> **Our Work in the Context of Previous Works**
>
> We thank the reviewer for pointing out further related work. While both NeRFlow and our approach use the concept of neural ODEs to differentiably integrate dynamics over time, we would like to point out that the approaches are still quite different. NeRFlow uses another coordinate-based representation to parametrize the dynamics in the scene. While this allows one to learn quite general dynamics of the scene, the result is not an interpretable representation, since the flow does not build on the notion of objects. On the other hand, we use a parametric model (that of course constrains the dynamic) but therefore allows us to extract interpretable physical parameters. We now cite this paper and also incorporated other suggestions of the reviewer.
>
> &nbsp;
>
> **Experimental Evaluation**
>
> We have included the example of two MNIST digits connected by a spring in the experimental section:
> 1. We show that our proposed method works for other physical systems, in this specific case two masses that are connected by a spring.
> 2. By comparing against the method proposed by Jaques et al. [1] we include another baseline as reference.
> 3. We show that the approach is able to handle two coupled dynamic objects.
> *[answer reproduced from our general response]*
>
> Overall, we believe that this experiment also helps in putting our work into the context of existing work.
>
> &nbsp;
>
> **Using the Physical Model as Prior Knowledge**
>
> We would like to emphasize again that we target the setting of physical parameter estimation. Since we are interested in the parameters of a specific setting, the problem is domain-specific by construction. Indeed, for another physical setting, we need to set up a new model for the dynamics, as we now also show in our newly included experiment. Nevertheless, we see this as a strength of our approach, since including the prior knowledge of the dynamics allows us to identify a parametric physical model from a single video, while still maintaining interpretability of the result. In principle, it is also possible to provide a collection of dynamical model candidates, and then to learn during inference which dynamical model explains the scene best. We consider this combination of physical knowledge with model identification an interesting direction for future work. We would like to highlight that the proposed model can handle physical phenomena that can be described by an ODE, which includes systems that do not conserve energy, e.g. the damped pendulum shown in the paper.
>
> &nbsp;
>
> [1] Miguel Jaques, Michael Burke, and Timothy M. Hospedales.  Physics-as-inverse-graphics:  Unsupervised physical parameter estimation from video. ICLR 2020.

---

> > ### Comment · Reviewer_s2K9 · 2021-11-29
> > **Reviewer Response**
> >
> > I thank the authors for responding to my comments. However, I remain unconvinced by the underlying emperical evaluation. The additional task with MNIST digits seems rather artificial to me. The system is only evaluated on a single real video, and is further constrained to a particular view of the pendulum. Finally, I think it would be good to demonstrate the system can do physical parameter estimation across multiple separate videos.

---

### Official Review · Reviewer_PZv8 · 2021-11-03

**Correctness:** 3
**Technical Novelty And Significance:** 2
**Empirical Novelty And Significance:** 2
**Recommendation:** 3
**Confidence:** 3

**Main Review:**

Strength:

- The proposed method deals with video prediction in physics-aware fashion by incorporating implicit physics models, which seems to be novel and innovative. The direction that this paper is heading towards is interesting.
- The presentation of the paper is clear and easy to follow.

Weakness:

- The whole framework is built upon an assumption that the video can be (near) perfectly decomposed into foreground objects and background, which is a very toy assumption and cannot be used in any complicated real video data.
- This paper assumes knowing the underlying physics dynamics (in this case pendulum), which is an unreasonable assumption. Other dynamics, if it exists in the video, will not be able to be modeled.
- The experiments are very weak. 1) only one physics dynamics model of the pendulum is shown; 2) For the pendulum, only one real video data is evaluated; 3) the other experiments are done on synthetically generated data, which are also very weak.
- How about there is more than one pendulum in the video? How about viewing the pendulum from another viewpoint such that the motion pattern is not a perfect "swing"? These are not shown in the paper at all.



**Summary Of The Paper:**

This paper proposed a model for learning physical parameters from video. The model takes in a video with a foreground object and background, and predicts the motion of the object in the video. The experiments are done on a set of videos of damped pendulums.

**Summary Of The Review:**

Though the direction that this paper is heading towards is interesting, the framework only works on a toy case. The experiments are very weak. This paper does not reach of the bar of the conference.

---

> ### Author Response · Authors · 2021-11-23
> **Response to Reviewer PZv8**
>
> We thank the reviewer for the valuable feedback, which will help us to further improve our work. We appreciate that the reviewer values the clear presentation of the paper and considers the proposed research direction as novel and interesting. In a response to all reviewers, we have briefly listed the main changes to the manuscript, which are also highlighted in blue in the PDF. In the following, we will address specific points raised in the review.
>
> &nbsp;
>
> **Decomposition into Foreground and Background & Non-planar Motion**
>
> We agree that the current approach will not be able to handle pendulum motions in a plane that is not near-parallel to the image plane. However, the proposed method can be extended to cover this case. More generally, we consider the extension to 3D scenes an exciting research perspective. By proposing a general framework that utilizes coordinate-based neural representations, which have been shown to work remarkably well in 3D, we have made an initial step in this direction. Therefore, we believe that this paper on its own already forms an important contribution, in particular since it establishes an elegant and versatile representation to learn physical parameters from a single video clip. Additionally, we note that the assumption of compositionality (foreground and background separation) is common and has also been used in prior work, e.g. Ost et al. [2]. Getting rid of this assumption is also a highly relevant (yet challenging) future direction.
>
> &nbsp;
>
> **Knowledge of the Underlying Physics Dynamics Model**
>
> We would like to emphasize that we specifically target the setting of physical parameter estimation from a single video. The fact that we are relying on a significantly reduced amount of data (compared to existing methods that use large training corpora) implies that in such a setting a stronger prior is necessary. The problem of modeling dynamics without assuming a model in advance is an exciting area on its own, and we agree that such approaches give more flexibility. Overall, there is a trade-off between including prior knowledge of the dynamics, and the amount of data required for the identification. Here, our approach is located at one end of the extremes, where we assume the full knowledge of the physical model, and in return can deal with only a single video of the scene.
>
> &nbsp;
>
> **Experimental Evaluation**
>
> We now include an additional experiment in section 4.1, which serves the following purposes:
> 1. We show that our proposed method works for other physical systems, in this specific case two masses that are connected by a spring.
> 2. By comparing against the method proposed by Jaques et al. [1] we include another baseline as reference.
> 3. We show that the approach is able to handle two coupled dynamic objects.
> *[answer reproduced from our general response]*
>
> &nbsp;
>
> [1] Jaques et al,  Physics-as-inverse-graphics: Unsupervised physical parameter estimation from video. ICLR 2020.
>
> [2] Ost et. al, Neural scene graphs for dynamic scenes. CVPR 2021.

---

### Author Response · Authors · 2021-11-23
**Response to All Reviewers: Summary of Changes**

We thank all reviewers for their helpful comments that enabled us to further improve the paper. We appreciate that the reviewers share our belief that physical parameter estimation is an interesting field of research and we are happy that our paper is considered to be innovative and clear to follow.

&nbsp;

Besides smaller changes that clarify minor points and sharpen our argumentation, our most significant change is that we conducted **an additional experiment which is presented in section 4.1** (highlighted in blue in the revised version). The newly added experiment serves multiple purposes:
1. We show that our proposed method works for **other physical systems**, in this specific case two masses that are connected by a spring.
2. By comparing against the method proposed by Jaques et al. [1] we **include another baseline** as reference.
3. We show that the approach is able to **handle two coupled dynamic objects**.

We address the comments of each reviewer in the respective answers. For the reviewers’ convenience and for self-containedness, we reproduce our answers at appropriate places.

&nbsp;

[1] Miguel Jaques, Michael Burke, and Timothy M. Hospedales.  Physics-as-inverse-graphics:  Unsupervised physical parameter estimation from video. ICLR 2020.

---

### Decision · Program_Chairs · 2022-01-20

**Decision:**

Reject

**Comment:**

The paper proposes a framework for learning the physical parameters of a physical system’s dynamics from a video. The model combines a differentiable neural ODE solver (NODE) with neural implicit representations through a local coordinate-based network which reconstruct the frames based on the ODE solution. Both the static background and the moving objects are modeled via implicit representations. The system being differentiable, it can be trained to recover the physical parameters and the initial conditions of the ODE. Experiments are performed on two toy problems (pendulum and masses that are connected by a spring).

The reviewers agree on the originality of the approach. They however all consider that the paper falls short to demonstrate the potential of the proposed approach because of limited experiments, limited ablation analyses and comparison with baselines. The authors added a new experiment during the rebuttal, but this was not found sufficient to change the reviewers’ opinion.